# Assessment of cardiovascular risk in a slum population in Kenya: use of World Health Organisation/International Society of Hypertension (WHO/ISH) risk prediction charts - secondary analyses of a household survey

Amoolya Vusirikala,[1] Frederick Wekesah,[2,3] Catherine Kyobutungi,[2] Oyinlola Oyebode[1]

¹University of Warwick Warwick Medical School, Coventry, UK
²African Population and Health Research Center, Nairobi, Kenya
³Julius Center for Health Sciences and Primary Care, University Medical Center Utrecht, Utrecht University, Utrecht, Netherlands

**Correspondence to**
Dr Amoolya Vusirikala;
amoolyav@gmail.com

## ABSTRACT

**Objectives** Although cardiovascular disease (CVD) is of growing importance in low- and middle-income countries (LMICs), there are conflicting views regarding CVD as a major public health problem for the urban poor, including those living in slums. We examine multivariable risk prediction in a slum population and assess the number of cardiovascular related deaths within 10 years of application of the tool.

**Setting** We use data from the Nairobi Urban Health and Demographic Surveillance System (NUHDSS) population (residents of two slum communities) between May 2008 and April 2009.

**Design** This is a secondary data analysis from a cross-sectional survey. We use the WHO/International Society of Hypertension (WHO/ISH) cardiovascular risk prediction tool to examine 10-year risk of major CVD events in a slum population. CVD deaths in the cohort, reported up until June 2018 and identified through verbal autopsy are also presented.

**Participants** 3063 men and women aged over 40 years with complete data for variables needed for the WHO/ISH risk prediction tool were eligible to take part.

**Results** The majority of study members (2895, 94.5%) were predicted to have 'low' risk (<10%) of a cardiovascular event over the next 10 years and just 51 (1.7%) to have 'high' CVD risk (≥20%). 91 CVD deaths were reported for the cohort up until June 2018. Of individuals classified as low risk, 74 (2.6%) were identified as having died of CVD. Nine (7.7%) individuals classified at 10% to 20% risk and eight (15.9%) classified at >20% were identified as dying of CVD.

**Conclusions** This study shows that there is a low risk profile of CVD in this slum population in Nairobi, Kenya, in comparison to results from application of multivariable risk prediction tools in other LMIC populations. This has implications for health service planning in these contexts.

## Strengths and limitations of this study

► To the best of our knowledge this is the first study to apply a multivariable risk prediction tool to a population in a slum or informal settlement.
► We were able to identify cardiovascular disease (CVD) deaths of participants occurring in the slum during the 10 years after risk prediction.
► We were unable to exclude individuals with previous myocardial infarction as information was not available from the survey.
► Applying the risk score chart to cross-sectional population data may have underestimated the total CVD risk, as data that are required for thorough evaluation of total risk (such as family history) was absent.

middle-income countries (LMICs).[1] Cardiovascular disease (CVD) is a key player in this epidemic, accounting for most NCD deaths, and studies of CVD in urban areas of LMICs have suggested that risk is growing.[2–4]

A large proportion of the world's urban population live in slums — neighbourhoods that are often informal, with poor housing and inadequate services.[5] There are conflicting views regarding CVD as a major public health problem for the urban poor, including those living in slums. An overview of health in slums found no synthesised evidence on CVD prevalence or the prevalence of CVD risk factors, while primary studies indicated that some CVD risk factors appear to be less prevalent among those living in slums than in their non-slum urban counterparts.[5] However, other primary studies carried out in urban LMICs have indicated that CVD risk is inversely associated with socioeconomic status, or that there is no strong socioeconomic gradient, which would suggest

## INTRODUCTION

Non-communicable diseases (NCDs) are the leading cause of death globally and have become the leading cause of death in low- and

that those living in slums had at least equivalent or higher risk than other urban residents.[6 7]

Conventionally, CVD risk prediction focused on the presence of certain individual risk factors (eg, elevated blood pressure or serum cholesterol), however the recognition of the multifactorial aetiology of CVD has led to a drastic shift away from the single risk factor approach toward a multivariable risk prediction approach. Taking into account the coexistence of multiple risk factors to determine CVD risk has been supported by much research that clearly demonstrates that the risk of a CVD event can differ among individuals with the same high levels of single risk factors due to the presence or absence of other risk factors.[8–10] Furthermore, studies have shown that identifying individuals at high CVD risk by adopting a total CVD risk assessment approach is more cost-effective method of CVD prevention especially in low resource settings.[11 12] Determining total CVD risk requires risk prediction tools. The WHO/International Society of Hypertension (WHO/ISH) developed risk score prediction charts for different WHO subregions for the purposes of enabling clinicians to quickly assess total CVD risk in their patients, but also allow for risk stratification of a population in a simple manner.[13]

However, there are no existing studies that have assessed multivariable risk prediction of CVD in a slum population. Therefore, the primary aim of this study is to apply the WHO/ISH risk prediction tool to a slum population in Nairobi, Kenya. In addition, we were able to assess the number of cardiovascular-related deaths occurring within the slum (but not non-fatal events, or fatal events occurring elsewhere) reported within 10 years of application of the tool, giving us some idea about the utility of the WHO/ISH tool in this population, and about the burden of CVD within the slum setting. These findings will inform plans for health service delivery in the context of urban poor settings.

## METHODS
### Study population
This study utilises data from a cross-sectional survey conducted by the African Population and Health Research Center (APHRC) within the Nairobi Urban Health and Demographic Surveillance System (NUHDSS) population between May 2008 and April 2009. The NUHDSS was setup to examine the long-term social, economical and health effects of two slum communities within the city of Nairobi, Korogocho and Viwandani.[14] This population-based survey utilised the sampling frame from the NUHDSS and a stratified, sampling strategy based on the WHO STEPwise protocol with a target of 250 respondents in each of the following strata: sex, age group (18 to 24, 25 to 30, 31 to 40, 41 to 50, 51 to 60 and 60+), and slum of residence (Korogocho and Viwandani). Data were collected from a total of 5470 individuals aged 18 years and above. Further details on the sampling frame and data collection methods are published elsewhere.[15] Men and women aged over the age of 40 years with complete data for variables needed for the

WHO/ISH risk prediction tool were eligible to take part in this secondary data analysis.

### Ethics
Operations of the NUHDSS are approved by the Government of Kenya, and the ethical responsibilities for its operations overseen by the Kenya Medical Research Institute (KEMRI). The CVD study was approved by the Scientific and Ethics Review unit of KEMRI (SERU NON-SSC 339). Participants were made aware that their participation in the study was voluntary, and individual informed consent was sought from all participants before their involvement in the study.

### Study tool
In order to assess the 10-year risk of fatal and non-fatal cardiovascular disease (namely myocardial infarction or stroke) for each participant in our sample, we used the WHO/ISH risk prediction charts for Africa sub-region (AFR E).[13] The charts are designed for those over 40 and those who do not have established coronary heart disease (CHD), stroke or other atherosclerotic disease. Therefore, our study sample excluded those <40 years of age and those with stroke. We were unable to identify and exclude those who had established CHD or other atherosclerotic disease as the information was not available from the survey.

The chart requires data on sex, age, systolic blood pressure, smoking status and presence or absence of diabetes mellitus. If total serum cholesterol is available, this is used in CVD risk prediction, however there is also an algorithm for use where no total serum cholesterol record is available, which we used to calculate risk for the study participants with missing cholesterol data. Studies have demonstrated high correlation between laboratory-based scores and non-laboratory-based scores for men and women.[16 17]

Following guidelines for using the WHO/ISH risk prediction tool variables were constructed as follows: smokers were considered as those who were current smokers at assessment or those who quit smoking within the last year before the assessment, presence of diabetes was defined as someone taking insulin or oral hypoglycaemic drugs or having a study-measured fasting glucose >7.0 mmol/L, systolic blood pressure was the average of three readings on the day of survey while study member was seated using OMRON M6 blood pressure machine, total cholesterol (mmol/l) was measured by taking capillary blood from fingertips using the Accu-Chek Glucose, Cholesterol and Triglycerides (GCT) monitors and test strips. The predicted risk falls into categories including: from <10% (low), 10 to <20% (moderate), 20 to <30% (high), 30 to <40% (very high) and 40% or more (highest).[18]

In addition, practice points accompany the WHO risk prediction charts and state that CVD risk may be elevated over that specified by the charts when certain factors are present. We were able to obtain the following CVD enhancing risk factors for our study members: raised triglycerides (>2.0 mmol/L), whether on hypertensive medication and presence of obesity defined according to

body mass index (weight in kg divided by the square of height in cm). The prevalence of these CVD risk elevating factors were tabulated by the different risk categories.

## Identifying cardiovascular related deaths within 10 years of application of the tool

To assess the cardiovascular-related deaths among the participants of this study, verbal autopsy data present for all deaths recorded between 2008 and June 2018 was obtained from APHRC. A record linkage was undertaken between the cross-sectional survey and the verbal autopsy data using a unique identifier present in both data sources.

Verbal autopsy interviews are conducted by experienced field interviewers with a 'credible respondent', usually a family member following identification of deaths during regular Demographic Surveillance Site (DSS) data collection.[19] A standardised questionnaire developed in conjunction with other International Network of Demographic Evaluation of Populations and Their Health (INDEPTH) sites is used and consists of open and closed questions focusing on events leading to the death and specific clinical signs and symptoms that the deceased had prior to their death. After several visits to a household, if no 'credible respondent' is identified, verbal autopsy is coded as missing, and no cause of death is recorded.

Cause of death is then generated using InterVA-4 software, which uses probabilistic models based on Bayes' theorem to interpret symptom and signs data from verbal autopsy questionnaires and determine possible causes of death. Detailed information of the InterVA model and how it was developed have been described in previous studies.[19–21] Those with a cardiovascular code recorded under the variable 'first broad cause of death' was used to define cardiovascular-related death in this study. Cardiovascular death included ischaemic heart disease, cerebrovascular disease, hypertensive diseases, pulmonary heart disease and diseases of pulmonary circulation and diseases of arteries, arterioles and capillaries. We additionally examined deaths reported due to diabetes mellitus because someone who had diabetes at the time of death may have died from a cardiovascular outcome, but credible family members may only have discussed the condition they were suffering prior to death.

Please note that the deaths recorded in this study were deaths only identified during the regular data collection rounds by the DSS team, it may be that more deaths occurred among participants of the original cross-sectional survey that were not identified (for example, the participant had moved before the death took place) and non-fatal CVD events were not captured at all. We were not able to link individuals in this data set to know if they were still resident in the NUHDSS in June 2018 — but a larger study sample, including this one, identified just 53/4290 (1.2%) had exited the NUHDSS between 2008 and 2018 (Frederick Wekesah — personal communication). The published rate of out migration between 2003 and 2012 was 22.5%.[14] For these reasons we can't be sure how well the tool predicted CVD events in this population but include these figures to add to knowledge about the burden of disease and give some indication of the tool's performance.

Patients and public were not involved in this study.

## Statistical analyses

Data analysis was conducted in Stata, V.14.0 (StataCorp LP, College Station, USA). Percentages were calculated for categorical variables. Sampling weights were applied where noted.

# RESULTS

## Description of sample

Of the 5470 survey participants, 2316 were excluded due to age (<40 years). Of the remaining 3154 participants aged over 40, 10 were excluded due to having a stroke. Eighty-one (2.6%) were excluded due to incomplete data for the variables required of the WHO/ISH risk prediction tool (missing data for smoking status (n=2) and for blood pressure (n=81)).

Characteristics of the 3063 participants included in the final sample are represented in table 1. The majority of participants included in the analyses were male (57.6%) and were between 40 to 60 years old, 88% of participants included in the analyses were non-smokers, 3% had diabetes and approximately 24% had blood pressure ≥140 mm Hg.

## Population distribution of CVD risk using WHO/ISH risk prediction chart and prevalence of CVD risk-enhancing factors

The majority of participants in this study sample had low (<10%) total 10 year CVD risk (2895, 94.5%) [table 2]. That is, they had less than 10% predicted chance of a fatal or non-fatal CVD event over the following 10 years. When CVD-risk enhancing factors were taken into account, 1963 participants (64.1%) had low (<10%) total 10 year CVD risk with no additional risk enhancing factors. In the weighted analysis, the percentage of people in the <10% risk group was 96.3% with the reduced risk profile due to correction of the over-sampling of older age groups. After applying the CVD-enhancing factors, the percentage of people in the <10% risk group reduced to 63.7% in the weighted analysis.

WHO also states risk from prediction charts alone can underestimate the risk in those with high blood pressure (≥160/100) or blood cholesterol ≥8. In our sample, 9.83% had raised blood pressure but only 0.07% of those who had a cholesterol blood test were raised above the specified level. Of those at <10% risk group, 131 participants had raised blood pressure above or equal to 160/100 and of those in this category that had a blood cholesterol test only two participants had a blood cholesterol ≥8.

## Cardiovascular related-deaths at different levels of predicted CVD risk based on chart alone

Following record linkage of verbal autopsy database and the respondents of the cross-sectional survey, 466 records were matched: that is 466 of the original cohort had a death reported up to June 2018 among the participants from the cross-sectional survey (n=5470). Among the

**Table 1** Characteristics of study participants

| | Total (n=3063) | Men (n=1765) | Women (n=1298) |
|---|---|---|---|
| **Age in years (n, %)** | | | |
| 40–49 | 1168 (38.13) | 595 (33.71) | 573 (44.14) |
| 50–59 | 1169 (38.17) | 770 (43.63) | 399 (30.74) |
| 60–69 | 493 (16.10) | 294 (16.66) | 199 (15.33) |
| ≥70 | 233 (7.61) | 106 (6.01) | 127 (9.78) |
| **BMI category (n, %)*** | | | |
| <30 kg/m2 | 2714 (89.39) | 1700 (96.98) | 1014 (79.03) |
| ≥30 kg/m2 | 322 (10.61) | 53 (3.02) | 269 (20.97) |
| **Smoking status (n, %)** | | | |
| Current | 375 (12.24) | 364 (20.62) | 11 (0.85) |
| Non-smoker | 2688 (87.76) | 1401 (79.38) | 1287 (99.15) |
| **Blood pressure (n, %)** | | | |
| <140 | 2312 (75.48) | 1382 (78.30) | 939 (72.34) |
| 140–159 | 441 (14.40) | 237 (13.43) | 204 (15.72) |
| 160–179 | 185 (6.04) | 101 (5.72) | 84 (6.47) |
| 180+ | 116 (3.79) | 45 (2.55) | 71 (5.47) |
| **Diabetes (n, %)** | | | |
| Absent | 2976 (97.16) | 1727 (97.85) | 1249 (96.22) |
| Present | 87 (2.84) | 38 (2.15) | 49 (3.78) |
| **Total cholesterol (n, %)** | | | |
| <5 | 2220 (72.48) | 1307 (74.05) | 913 (70.33) |
| 5–5.9 | 516 (16.84) | 281 (15.92) | 235 (18.10) |
| 6–6.9 | 74 (2.42) | 32 (1.81) | 42 (3.24) |
| 7–7.9 | 10 (0.33) | 5 (0.28) | 5 (0.39) |
| 8+ | 2 (0.07) | 1 (0.06) | 1 (0.08) |
| Cholesterol missing | 241 (7.87) | 139 (7.88) | 102 (7.86) |

*Realistic BMI values only available for 3036 participants of our total sample size of 3063.
BMI, body mass index.

3063 participants included in this study, 410 deaths were recorded with 91 deaths specifically related to CVD (3% of the study population), while 34 were classified as indeterminate and in 34 further cases, a verbal autopsy was not performed. Cardiovascular related cause of death was assigned to 74 (2.6%) of individuals classified as low risk (<10% predicted chance of a fatal or non-fatal CVD event). Nine (7.7%) of individuals classified at 10% to 20% risk of a fatal or non-fatal CVD event were determined to have died from CVD, and eight (15.9%) of those at high risk (≥20%) (table 3). Out of 336 deaths in <10% group, 87 individuals who had died had one or more CVD-enhancing factors, 18 of these deaths were due to CVD risk and 2 deaths were due to diabetes.

## DISCUSSION

Of the 3063 study members aged over 40, the majority (94.5%) were predicted to have a less than 10% chance of experiencing a cardiovascular event over the next 10 years and just 1.7% having a 'high' CVD risk (≥20%). When weighted to be representative of all adults aged over 40 living in the slum 96.3% were predicted to fall in the lowest risk group and just 1.2% have a 'high' CVD risk (≥20%). This is a low risk population profile in comparison to results from application of the multivariable risk prediction tools in other populations. Studies conducted specifically among urban dwellers in LMIC countries such as Malaysia[22] and Sri Lanka[23] have found 20.5% and 8.2% individuals were at high risk (≥20%) of having a future CVD event, respectively. Furthermore, the proportion of individuals in our study shown to be at low risk of a CVD event over 10 years (<10% risk) was higher than that of studies who used the WHO/ISH risk prediction charts carried out in rural Nepal (86.4%),[24] rural South India (83%)[25] and rural Bangladesh (81.3%).[26] Mendis et al reported total 10 year CVD risk in defined geographical areas of seven countries including both urban and rural populations, but only two countries had a higher percentage of individuals classified as

**Table 2** Proportion of population % at different level of CVD risk & prevalence of CVD risk enhancing factors

| 10-year CVD risk | <10% Low | 10% to <20% Moderate | 20% to <30% High risk | 30% to <40% Very high risk | ≥40% Highest risk |
|---|---|---|---|---|---|
| **Total** | | | | | |
| CVD risk-prediction from WHO/ISH chart alone (n, %) | 2895 (94.5) | 117 (3.8) | 31 (1.0) | 14 (0.5) | 6 (0.2) |
| CVD risk-prediction from WHO/ISH chart alone (%) (weighted) | (96.3) | (2.5) | (0.8) | (0.3) | (0.1) |
| One or more CVD enhancing factors (n, % of risk category) | 932 (32.2) | 62 (53.0) | 20 (64.5) | 3 (21.4) | 5 (83.3) |
| Obesity (BMI ≥30) (n, % of risk category*) | 292 (10.2) | 21 (18.3) | 9 (29.0) | 0 (0) | 0 (0) |
| On anti-hypertensive medication (n, % of risk category) | 75 (2.6) | 20 (17.1) | 6 (19.4) | 1 (7.1) | 2 (33.3) |
| High triglycerides (n, % risk category†) | 714 (27.6) | 46 (42.6) | 12 (44.4) | 3 (25.0) | 4 (80.0) |
| **Male** | | | | | |
| CVD risk-prediction from WHO/ISH chart alone (n, %) | 1679 (95.1) | 59 (3.3) | 12 (0.7) | 11 (0.6) | 4 (0.2) |
| CVD risk-prediction from WHO/ISH chart alone (%) (weighted) | (95.2) | (3.5) | (1.1) | (0.2) | (0.1) |
| One or more CVD enhancing factors (n, % of risk category) | 443 (26.4) | 21 (35.6) | 6 (50.0) | 2 (18.1) | 3 (75.0) |
| Obesity (BMI ≥30) (n, % of risk category*) | 49 (2.9) | 2 (3.4) | 2 (16.7) | 0 (0.0) | 0 (0.0) |
| On anti-hypertensive medication (n, % of risk category) | 22 (1.3) | 6 (10.2) | 1 (8.3) | 0 (0.0) | 0 (0.0) |
| High triglycerides (n, % risk category†) | 400 (26.6) | 16 (30.2) | 4 (44.4) | 2 (22.2) | 3 (75.0) |
| **Female** | | | | | |
| CVD risk-prediction from WHO/ISH chart alone (n, %) | 1216 (93.7) | 58 (4.5) | 19 (1.5) | 3 (0.2) | 2 (0.2) |
| CVD risk-prediction from WHO/ISH chart alone (%) (weighted) | (96.8) | (2.1) | (0.7) | (0.3) | (0.2) |
| One or more CVD enhancing factors (n, % of risk category) | 489 (40.2) | 41 (70.7) | 14 (73.7) | 1 (33.3) | 2 (100) |
| Obesity (BMI ≥30) (n, % of risk category*) | 243 (20.2) | 19 (33.9) | 7 (36.8) | 0 (0.0) | 0 (0.0) |
| On anti-hypertensive medication (n, % of risk category) | 53 (4.4) | 14 (24.1) | 5 (26.3) | 1 (33.3) | 2 (100) |
| High triglycerides (n, % risk category†) | 314 (28.9) | 30 (54.5) | 8 (44.4) | 1 (33.3) | 1 (100) |

*Realistic BMI values only available for 3036 participants of our total sample size of 3063.
†High triglycerides were recorded in 2739 participants of our total sample size of 3063.
BMI, body mass index; CVD, cardiovascular disease; WHO/ISH, WHO/International Society of Hypertension.

**Table 3** Total deaths and deaths due to cardiovascular disease recorded up to June 2018 at different levels of predicted CVD risk (based on chart alone)

| 10-year CVD risk | <10% Low | 10%–20% Moderate | 20%–30% High risk | 30%–40% Very high risk | ≥40% Highest risk |
|---|---|---|---|---|---|
| **Total** | | | | | |
| Deaths recorded (n, %) | 336 (11.6) | 25 (21.4) | 11 (35.5) | 7 (50.0) | 1 (16.7) |
| Broad first cause of death: cardiovascular (n, %) | 74 (2.6) | 9 (7.7) | 5 (16.1) | 2 (14.3) | 1 (16.7) |
| Broad first cause of death: diabetes Mellitus (n, %) | 8 (0.3) | 0 (0.0) | 0 (0.0) | 0 (0.0) | 0 (0.0) |
| Indeterminate cause (n, %) | 30 (1.0) | 4 (3.4) | 0 (0.0) | 0 (0.0) | 0 (0.0) |
| VA not done (n, %) | 33 (1.1) | 1 (0.9) | 0 (0.0) | 0 (0.0) | 0 (0.0) |
| **Male** | | | | | |
| Deaths recorded (n, %) | 199 (11.9) | 10 (16.9) | 5 (41.7) | 4 (36.4) | 0 (0.0) |
| Broad first cause of death: cardiovascular (n, %) | 35 (2.1) | 3 (5.1) | 2 (16.7) | 1 (9.1) | 0 (0.0) |
| Broad first cause of death: diabetes Mellitus (n, %) | 3 (0.2) | 0 (0.0) | 0 (0.0) | 0 (0.0) | 0 (0.0) |
| Indeterminate cause (n, %) | 16 (1.0) | 1 (1.7) | 0 (0.0) | 0 (0.0) | 0 (0.0) |
| VA not done (n, %) | 22 (1.3) | 1 (1.7) | 0 (0.0) | 0 (0.0) | 0 (0.0) |
| **Female** | | | | | |
| Deaths recorded (n, %) | 137 (11.3) | 15 (25.9) | 6 (31.6) | 3 (100) | 1 (50.0) |
| Broad first cause of death: cardiovascular (n, %) | 36 (3.1) | 6 (10.3) | 3 (15.8) | 1 (33.3) | 1 (50.0) |
| Broad first cause of death: diabetes mellitus (n, %) | 5 (0.4) | 0 (0.0) | 0 (0.0) | 0 (0.0) | 0 (0.0) |
| Indeterminate cause (n, %) | 141 (11.6) | 3 (5.2) | 0 (0.0) | 0 (0.0) | 0 (0.0) |
| VA not done (n, %) | 11 (0.9) | 0 (0.0) | 0 (0.0) | 0 (0.0) | 0 (0.0) |

CVD, cardiovascular disease; VA, verbal autopsy.

low risk in comparison to our study (lower: Iran (93.9%), Cuba (89.7%), Nigeria (86.0%), Georgia (83.1%), Pakistan (79.2%); similar: China (96.1%) and Sri Lanka (94.9%)).[12] However, it is important to note, the proportion of individuals estimated to have low (<10%) total CVD risk substantially decreased in our study, when risk-elevating factors stated in practice points accompanying WHO/ISH charts (raised triglycerides, obesity and anti-hypertensive medication) were added to the CVD risk assessment of the population.

CVD deaths occurring in our study population within the slum, reflected risk-categories assigned by the WHO/ISH tool (bearing in mind that additional deaths may have occurred in study members outside the slum and that non-fatal events were not recorded). Taken as a whole, it appears from this data that health services geared towards CVD treatment may be less of a priority in slum settings in Kenya, or potentially in sub-Saharan Africa, than in the wider urban areas of LMIC cities. An important reason for this may be the age-structure of the slum population, which is very young. However, given the large percentage of CVD-enhancing factors in this population, it could be that the future burden (when this population gets older) will be

significant. The signal here could be that CVD prevention is more of a priority here than treatment.

Certain limitations of this work need to be considered when interpreting the findings. First, we were unable to exclude individuals with previous myocardial infarction as information was not available from the survey. However, if we failed to identify significant numbers with a previous myocardial infarction, the remaining population (once these individuals had been excluded) would have likely had an even more extremely low risk profile for CVD. Second, applying the risk score chart to cross-sectional population data may have underestimated the total CVD risk, as data that are required for thorough evaluation of total risk such as family history or even history of relevant current diseases (the obvious example being myocardial infarction) and treatments, were not present in the data. Third, there are some deviations in our methods from the instructions of how the WHO/ISH charts should be used: systolic blood pressure was measured three times on 1 day, rather than twice at two different time points, which could increase the risk that some of the participants experienced white-coat hypertension; we defined someone as having diabetes if they were taking insulin or oral hypoglycaemic drugs or

if their fasting plasma glucose concentration was about 7.0 mmol/L on one occasion (not on two separate occasions as recommended). Finally, where we used cholesterol readings — these were also from one time point, rather than two as recommended.

Finally, it is a regret that we don't have data on all possible fatal CVD events (for example in those who have moved from the study site and are therefore not followed up in the NUHDSS) or non-fatal events that have occurred in the 10 years since the risk data was collected in order to validate the WHO/ISH tool in this setting.

Despite these limitations, our study uses rare data to provide a good estimate of total 10 year CVD risk among a marginalised population in an urban poor setting in sub-Saharan Africa. To the best of our knowledge this is the first study to apply a multivariable risk prediction tool to a population in a slum or informal settlement and to assess the number of cardiovascular related deaths within 10 years of application of the tool. This study shows that there is a low risk profile of CVD in this slum population in Nairobi, Kenya, and that the WHO/ISH tool does differentiate groups at increasing risk of CVD mortality. This has implications for planning of health service delivery in slums.

**Contributors** OO conceived the study idea. OO, AV and FW contributed to the analysis plan. AV conducted the analyses. FW and CK provided advice on using the data. AV and OO wrote the first draft of the manuscript. All authors contributed to the final manuscript.

**Funding** OO and CK are supported by the National Institute for Health Research (NIHR) Research Unit on Improving Health in Slums. The original data on which the study is based was collected as part of research funded by the Wellcome Trust UK - Grant Number WT092775MA. This research did not receive any specific grant from funding agencies in the public, commercial or not-for-profit sectors. This paper presented independent research and the views expressed are those of the authors and not necessarily those of the NHS, the NIHR or the Department of Health.

**Competing interests** None declared.

**Patient consent for publication** Not required.

**Provenance and peer review** Not commissioned; externally peer reviewed.

**Data availability statement** Data may be obtained from a third party and are not publicly available. All data relevant to the study are included in the article or uploaded as supplementary information.

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
