## [Reviewer comments · BMJ Open]

ARTICLE DETAILS

TITLE (PROVISIONAL)	Assessment of cardiovascular risk in a slum population in Kenya: use of World Health Organization/International Society of Hypertension (WHO/ISH) risk prediction charts- Secondary analyses of a Household Survey
AUTHORS	Vusirikala, Amoolya; Wekesah, Frederick; Kyobutungi, Catherine; Oyebode, Oyinlola

VERSION 1 - REVIEW

REVIEWER	Rosario Caruso IRCCS Policlinico San Donato
REVIEW RETURNED	11-Mar-2019

GENERAL COMMENTS	Thank you for giving me the possibility to review this interesting study. Overall, the manuscript is well-written and the investigation provides interesting insights, covering an actual gap in research. I do not understand why no inferential analysis was performed to complete the clear descriptive analysis: e.g. comparison between the sub-groups of men and women related to the investigated characteristics (further, it could be interesting if you provide a rationale on this stratification); the differences of 10-year CVD risk/deaths (table 2 and 3) and CVD enhancing factors between the sub-groups belonging to the different categories of predicted risk. I also suggest to better define a conclusion paragraph. Some minor issues: Citations: Please, place superscript numerals outside periods and commas, inside colons and semicolons, as per AMA style Paragraph pag 6 (rows 7-21): it is not needed to capitalize the word after a semicolon.
--

REVIEWER	Nain-Feng Chu National Defense Medical Center, Taiwan
REVIEW RETURNED	27-Mar-2019

GENERAL COMMENTS	This is an interested manuscript to evaluate the CVD risk score in relationship to 10-year CVD mortality among slum population in Kenya. The authors found that the CVD mortality was 2.6%, 7.7% and 15.9% for low, middle and high CVD risk score population.
--

	However, there are some points should be addressed in this manuscript.  1. It would be more interpretable to calculate the OR or RR of CVD mortality for different CVD ore population after adjusting for the potential confounders – such as gender or age 2. It is very interested that the characteristics of study population showed the women had more prevalent of obesity, hypertension and diabetes than the men. 3. The history of medications, such as hypertension and diabetes should be considered in the following analyses. 4. The dead due to total and CVD could with gender-specification.
--	---

REVIEWER	Pär Andersson White Linköping University, Sweden
REVIEW RETURNED	25-Apr-2019

GENERAL COMMENTS	An interesting study in a hard to reach population! However I find some things that needs to be addressed. 3. Is the study design appropriate to answer the research question? There are some deviations from the instructions of how the WHO/ISH charts should be used that needs to be discussed. Systolic blood pressure should be measured at two separate time point; using only one time point increase the risk of White-coat hypertension and this should be discussed. I believe you used fasting glucose >7.0 as a diagnostic for diabetes and not OGTT = oral glucose tolerance test. Cholesterol should be measured at 2 time points if dry chemistry is used. These deviations from the recommended application should be discussed as potential sources of error. 4. Are research ethics (e.g. participant consent, ethics approval) addressed appropriately? Please comment the ethical considerations of NUDHSS, I only find a reference 14 which in turn uses a reference to describe NUDHSS. 7. If statistics are used are they appropriate and described fully? I find no explanation on whether weights were used in this sample (as it is in ref 14). If there were no significant difference in the proportions of age groups to the population then please state so. Table 2 Are these figures weighted data to the size of the age groups of the population? As age is very important for CVD risk then if the sample is older than the population the risk might be an overestimation. 11. Are the discussion and conclusions justified by the results? The result of this study indicates that CVD risk in slum areas of Nairobi is low, however no figures on cardiovascular health in the rest of the city is presented. Thus one cannot conclude whether the low CVD risk is a specific feature of slum areas or the city of Nairobi. A number of studies on cardiovascular risks in different populations are discussed, most of them showing considerably
---

	higher prevalence of middle and high CVD risk groups. The article from Rio de Janeiro described in the introduction of this article shows, although using other methods than WHO/ISH, a prevalence of 42.2 % in males 62.5 % in females of increased risk with no significant difference between slum residents and others. Thus generalizability of these results should not be overstated, a reasonable statement would be then that CVD risk does not seem to be important in slum areas in Kenya and possibly not in similar cities of sub-Saharan African countries, but is likely to be higher in middle income countries as shown in previous studies (Rio de Janeiro). 12. Are the study limitations discussed adequately? It is difficult to know whether the WHO/ISH provide a good estimate of 10 year CVD risk in slum areas without knowing the incidence of myocardial infarctions and stroke (only mortality). The study show however that WHO/ISH prediction charts seems to predict some of the cardiovascular mortality in slum areas of Kenya. As cardiovascular risk is dependent on age, a discussion on the age pyramid of the slum area studied would be interesting. If the population is very young, which seems to be the case with only 7.6 % of those above 40 being older than 70 then that would in turn explain a low cardiovascular risk. Also it would be interesting to know the whole age pyramid of the area, if for example 80 % are below 40 (approx. the case in Kenya according to WHO) then the number at risk for CVD would be very low.
--	---

REVIEWER	Eleanor L Axson National Heart and Lung Institute Imperial College London United Kingdom
REVIEW RETURNED	20-May-2019

GENERAL COMMENTS	This is an interesting paper; however, there are a few things missing that would strengthen this paper greatly. Please consider the following suggestions: Please discuss ethical considerations/approvals. Please specify that you are only looking for CV deaths due to MI or stroke, not other CV causes. Also, you mention risk for non-fatal CV outcomes (MI and stroke) as outcomes in the first sentence of your methods; however, you state that these were not available in your introduction and your limitations – for clarity perhaps remove this from your methods. Please provide a reasoning for defining low risk as <10%, high risk as >20%. The definition appears to vary by study. For example, Mendis et al. categorise high risk as >30%. The reasoning behind including diabetes deaths is confusing. If you are interested MI and stroke deaths (as those are the risks predicted by the WHO/ISH tool), then why would you consider persons who died from diabetes as possible CV deaths? Why is a
--

	credible family member discussing diabetes prior to death considered questionable, while a credible family member discussing chest pain prior to death is not considered questionable? The significant difference in classification between the risk prediction WHO/ISH chart alone and the chart + enhancing factors (94.5% low vs 64.1% low) is very interesting and, I believe, warrants more discussion. Do any other studies look at the categorisation of risk based on the chart + enhancing factors (or some of the factors) groupings? Why does WHO/ISH only put the enhancing factors in the 'practice points' and not the final chart? Why or why not use the chart + enhancing groupings? Could it be recommended that studies use the chart + enhancing factors groupings? What were the number of deaths in each risk category (low, intermediate, high) based on the chart + enhancing factors groupings (include these in Table 3)? Does this categorisation lead to a different conclusion as to the necessity of CVD interventions in these communities? In your discussion, you briefly mention that 'CVD prevention and treatment may be less of a priority in slum settings than in the wider urban areas of LMIC cities'; however, you do not provide any further information. What are 'wider' urban areas? What do studies of these wider urban areas suggest about CVD risk and mortality? How do these estimates compare with your findings? How do the population demographics and risk factors compare between wider urban areas and your population (e.g. older, wealthier, obesity, etc. = more likely to develop CVD)? Additionally, the following minor revisions should be made:  1. Please cite the WHO/ISH charts, guidelines, and practice points. 2. Please include categorisation as 'low', 'intermediate', and 'high' risk in your Tables 2 & 3. 3. Please include totals (prediction form + enhancing factors) at the bottom of Table 2; such that the number in the second column would be 1963 (64.1). 4. Citations number 22 and 25 are missing journal information.
--	---

VERSION 1 – AUTHOR RESPONSE

Reviewer 1	
Thank you for giving me the possibility to review this interesting study. Overall, the manuscript is well-written and the investigation provides interesting insights, covering an actual gap in research	Thank you for your kind consideration of our article.
I do not understand why no inferential analysis was performed to complete the clear descriptive analysis: e.g. comparison between the sub-groups of men and women related to the investigated	The aim of the study was to examine the risk profile of people living in the slums. The analyses we report match these aims.

characteristics (further, it could be interesting if you provide a rationale on this stratification)	Are you asking why we stratified by sex in Table 1? This is just to give the reader more information about where the burden of cardiovascular risk factors lie.
The differences of 10-year CVD risk/deaths (table 2 and 3) and CVD enhancing factors between the sub-groups belonging to the different categories of predicted risk.	We have presented the number of individuals in each risk category that had one or more CVD enhancing factors (obesity, anti-BP meds and/or high trig) as “One of more CVD enhancing factors” now in Table 2. We haven’t put these people in a new category as it is not clear which category they should move up into (i.e.: not all enhancing factors are necessarily equivalent e.g.: the extra risk accrued by being obese, compared with the extra risk accrued from having high triglycerides) and whether these are additive if a person has more than one is also not clear. So the exact line you wanted in the table does not appear, but the new information allows you to estimate it, if for example you want to move all the people with one or more CVD enhancing factor from <10% up one category to 10-20% (but note this is not what is described in the WHO manual).
I also suggest to better define a conclusion paragraph.	We have added detail to our conclusion paragraph to better define the conclusion.
Place superscript numerals outside periods and commas, inside colons and semicolons, as per AMA style Paragraph pag 6 (rows 7-21): it is not needed to capitalize the word after a semicolon.	Thanks for highlighting, we have amended accordingly.
Reviewer 2	
This is an interested manuscript to evaluate the CVD risk score in relationship to 10-year CVD mortality among slum population in Kenya. The authors found that the CVD mortality was 2.6%, 7.7% and 15.9% for low, middle and high CVD risk score population	Thank you for your comments which is very much appreciated.
It would be more interpretable to calculate the OR or RR of CVD mortality for different CVD ore population after adjusting for the potential confounders – such as gender or age	The WHO/ISH tool that we have used predicts risk of a cardiovascular event over 10 years in categories as described (<10%, 10-20%, 20-30% etc.). Other risk prediction tools give a similar output (eg: Framingham, qrisk2). Here we show the n number and percentage of people who have died due to CVD in each category in order to facilitate the comparison between the percentage of people who died in each category with their predicted risk. We think this makes more sense than calculating ORs or RRs. (and please note that gender and age are integral to the tool- so they are effectively adjusted for).
It is very interested that the characteristics of study population showed the women had more	Thank you for your interest in our findings.

prevalent of obesity, hypertension and diabetes than the men.	
The history of medications, such as hypertension and diabetes should be considered in the following analyses.	We believe that this comment was sufficiently addressed in the submitted manuscript. However, apologies if we have misunderstood your point. Based on our interpretation of your comment: We state in the methods “the presence of diabetes was defined as someone taking insulin or oral hypoglycaemic drugs or having a study-measured fasting glucose > 7.0 mmol/l.” This is an integral part of the WHO/ISH tool (i.e.: the history of diabetes medication is considered in the analysis) “on hypertensive medication” is one of the CVD enhancing factors within this tool, and we present data on who is on hypertensive medication in Table 2.
The dead due to total and CVD could with gender-specification.	We have added to Table 2 & 3 and stratified by male and female as well as total.
Reviewer 3	
An interesting study in a hard to reach population!	Thanks for your comments which have improved the quality of our manuscript.
There are some deviations from the instructions of how the WHO/ISH charts should be used that needs to be discussed. Systolic blood pressure should be measured at two separate time point; using only one time point increase the risk of White-coat hypertension and this should be discussed. I believe you used fasting glucose >7.0 as a diagnostic for diabetes and not OGTT = oral glucose tolerance test. Cholesterol should be measured at 2 time points if dry chemistry is used. These deviations from the recommended application should be discussed as potential sources of error.	We have added this to the discussion: “there are some deviations in our methods from the instructions of how the WHO/ISH charts should be used: systolic blood pressure was measured three times on one day, rather than twice at two different time points, which could increase the risk that some of the participants experienced white-coat hypertension; cholesterol (although optional) should be measured at two time points, we defined someone as have diabetes if they were taking insulin or oral hypoglycaemic drugs or if their fasting plasma glucose concentration was about 7.0mmol/l on one occasion (not on two separate occasions as recommended). Finally, where we used cholesterol readings- these were also from one time point, rather than two as recommended.”
Please comment the ethical considerations of NUDHSS, I only find a reference 14 which in turn uses a reference to describe NUDHSS.	We have added this to the methods. Please see our response to the editorial requests above.
I find no explanation on whether weights were used in this sample (as it is in ref 14). If there were no significant difference in the proportions of age groups to the population then please state so.	Thank you for pointing this out. We have re-done analyses using the sampling weights which were available to us, and now report these in Tables 2 and 3. We have added the following to the discussion: “When weighted to be representative of all adults aged over 40 living in the slum 96.3% were predicted to fall in the

	lowest risk group and just 1.2% have a “high” CVD risk ($\geq 20\%$).” We have added a line to the methods under “statistical analyses” “Sampling weights were applied where noted.”
The result of this study indicates that CVD risk in slum areas of Nairobi is low, however no figures on cardiovascular health in the rest of the city is presented. Thus one cannot conclude whether the low CVD risk is a specific feature of slum areas or the city of Nairobi.	True, it is hard to compare here, since we only examined slums. We have added to our discussion a paragraph addressing this point, and the one below and your later comment on the age pyramid: “Taken in the whole it appears from this data that health services geared towards CVD treatment may be less of a priority in slum settings in Kenya, or potentially in sub-Saharan Africa, than in the wider urban areas of LMIC cities. An important reason for this may be the age-structure of the slum population, which is very young. However, given the large percentage of CVD-enhancing factors in this population, it could be that the future burden (when this population gets older) will be significant. The signal here could be that CVD prevention is more of a priority here than treatment.”
A number of studies on cardiovascular risks in different populations are discussed, most of them showing considerably higher prevalence of middle and high CVD risk groups. The article from Rio de Janeiro described in the introduction of this article shows, although using other methods than WHO/ISH, a prevalence of 42.2 % in males 62.5 % in females of increased risk with no significant difference between slum residents and others. Thus generalizability of these results should not be overstated, a reasonable statement would be then that CVD risk does not seem to be important in slum areas in Kenya and possibly not in similar cities of sub-Saharan African countries, but is likely to be higher in middle income countries as shown in previous studies (Rio de Janeiro).	See above.
Study limitations: It is difficult to know whether the WHO/ISH provide a good estimate	Agreed. We have stated in our discussion:

of 10 year CVD risk in slum areas without knowing the incidence of myocardial infarctions and stroke (only mortality). The study show however that WHO/ISH prediction charts seems to predict some of the cardiovascular mortality in slum areas of Kenya.	“Finally, it is a regret that we don’t have data on all possible fatal CVD events (for example in those who have moved from the study site and are therefore not followed up in the NUHDSS) or non-fatal events that have occurred in the 10 year since the risk data was collected in order to validate the WHO/ISH tool in this setting.” We have added to the final paragraph: “This study shows there is a low risk profile of CVD in this slum population in Nairobi, Kenya and that the WHO/ISH tool does differentiate groups at increasing risk of CVD mortality.”
As cardiovascular risk is dependent on age, a discussion on the age pyramid of the slum area studied would be interesting. If the population is very young, which seems to be the case with only 7.6 % of those above 40 being older than 70 then that would in turn explain a low cardiovascular risk. Also it would be interesting to know the whole age pyramid of the area, if for example 80 % are below 40 (approx. the case in Kenya according to WHO) then the number at risk for CVD would be very low.	We have added the following to the discussion: “Taken in the whole it appears from this data that health services geared towards CVD treatment may be less of a priority in slum settings in Kenya, or potentially in sub-Saharan Africa, than in the wider urban areas of LMIC cities. An important reason for this may be the age-structure of the slum population, which is very young. However, given the large percentage of CVD-enhancing factors in this population, it could be that the future burden (when this population gets older) will be significant. The signal here could be that CVD prevention is more of a priority here than treatment.”
Reviewer 4	
This is an interesting paper; however, there are a few things missing that would strengthen this paper greatly.	Thanks for your comments which have improved the quality of our manuscript.
Please discuss ethical considerations/approvals.	Completed, please see response to editorial comment.
Please specify that you are only looking for CV deaths due to MI or stroke, not other CV causes.	We have amended the methods to state: “CV death included ischaemic heart disease, cerebrovascular disease, hypertensive diseases, pulmonary heart disease and diseases of pulmonary circulation, and diseases of arteries, arterioles and capillaries.”
You mention risk for non-fatal CV outcomes (MI and stroke) as outcomes in the first sentence of your methods; however, you state that these were not available in your introduction and your limitations – for clarity perhaps remove this from your methods.	The WHO/ISH risk prediction tool predicts risk of fatal and non-fatal cardiovascular disease events. This is what is being described in that sentence under the “study tool” subheading within the methods section. This is the risk presented throughout the paper (i.e.: the <10% group have a <10% risk of a fatal or non-fatal CVD event). In the follow-up we were only able to look at CV deaths- which is what is explained in the introduction and discussion.

	To make this clearer we have added this to the Results section: “The majority of participants in this study sample had low (<10%) total 10-year CVD risk (2895, 94.5%) [Table 2]. That is they had less than 10% predicted chance of a fatal or non-fatal CVD event over the following 10 years.” “Cardiovascular related cause of death was assigned to 74 (2.6%) of individuals classified as low risk (<10% predicted chance of a fatal or non-fatal CVD event). Nine (7.7%) of individuals classified at 10-20% risk of a fatal or non-fatal CVD event were determined to have died from CVD, and 8 (15.9%) of those at high risk (\geq20%) [Table 3].”
Please provide a reasoning for defining low risk as <10%, high risk as >20%. The definition appears to vary by study. For example, Mendis et al. categorise high risk as >30%.	Prevention of cardiovascular disease: guideline for assessment and management of cardiovascular risk by WHO define low risk as <10% and 20-30% as high risk, >30% as very high risk. We used <10%, 10-20%, 20-30%, 30-40% and \geq40% risk categories to give readers more information across the spectrum of risk.
The reasoning behind including diabetes deaths is confusing. If you are interested MI and stroke deaths (as those are the risks predicted by the WHO/ISH tool), then why would you consider persons who died from diabetes as possible CV deaths?	We have presented the deaths due to diabetes separately, so you can examine our results with or without these. We think they are worth including in the manuscript for the reason given: that someone with diabetes may have died from a cardiovascular cause, but that when asked, the respondent may have said that the person died due to their diabetes because they had been known to have had diabetes for some time and this is what the person attributes their death to. We have not made any changes on the basis of this comment, but there is something specific you would like us to add, please let us know.
Why is a credible family member discussing diabetes prior to death considered questionable, while a credible family member discussing chest pain prior to death is not considered questionable?	I think you are asking again, why we included diabetes deaths? This is because diabetes is a long-term condition and it is possible that a respondent attributes the death of a family member to this condition, however, they may ultimately have died of a CVD event. We have included the deaths from diabetes for this reason, but we haven't amalgamated them with the CV deaths so you can examine the figures separately or together.
The significant difference in classification between the risk prediction WHO/ISH chart alone and the chart + enhancing factors (94.5% low vs 64.1% low) is very interesting and, I believe, warrants more discussion. Do any other studies look at the categorisation of risk based on the chart + enhancing	It is interesting. On reflection it seems possible that some of this is due to the young age-profile of the slum which means the 10-year predicted risk is low (and the number of deaths are low) however- actually the CVD risk enhancing factors are high, suggesting a future problem... We have tried to address this in the discussion i.e.: while still suggesting that treatment is low priority, prevention is probably more important (see response to your comment below).

factors (or some of the factors) groupings? Why does WHO/ISH only put the enhancing factors in the 'practice points' and not the final chart? Why or why not use the chart + enhancing groupings? Could it be recommended that studies use the chart + enhancing factors groupings? What were the number of deaths in each risk category (low, intermediate, high) based on the chart + enhancing factors groupings (include these in Table 3)? Does this categorisation lead to a different conclusion as to the necessity of CVD interventions in these communities?	Also please see our response to your later query about CVD-enhancing factors (i.e.: that we don't know exactly how much of a risk increase each one might add). The WHO/ISH is supposed to be for use in low resource settings so I think the idea was to create a risk prediction tool for the small set of data that might be easy and cheap to collect (+/- cholesterol results as described in our methods). The enhancing factors aren't part of the equation but just suggest a higher than predicted risk in an individual.
In your discussion, you briefly mention that 'CVD prevention and treatment may be less of a priority in slum settings than in the wider urban areas of LMIC cities'; however, you do not provide any further information. What are 'wider' urban areas? What do studies of these wider urban areas suggest about CVD risk and mortality? How do these estimates compare with your findings? How do the population demographics and risk factors compare between wider urban areas and your population (e.g. older, wealthier, obesity, etc. = more likely to develop CVD)?	Thank you- we have amended the discussion as follows: "Taken in the whole it appears from this data that health services geared towards CVD treatment may be less of a priority in slum settings in Kenya, or potentially in sub-Saharan Africa, than in the wider urban areas of LMIC cities. An important reason for this may be the age-structure of the slum population, which is very young. However, given the large percentage of CVD-enhancing factors in this population, it could be that the future burden (when this population gets older) will be significant. The signal here could be that CVD prevention is more of a priority here than treatment."
Please cite the WHO/ISH charts, guidelines, and practice points.	We have cited both charts, practice points and guidelines.
Please include categorisation as 'low', 'intermediate', and 'high' risk in your Tables 2 & 3.	We have added categorisation in Table 2 & 3.
Please include totals (prediction form + enhancing factors) at the bottom of Table 2; such that the number in the second column would be 1963 (64.1).	We have presented the number of individuals in each risk category that had one or more CVD enhancing factors (obesity, anti-BP meds and/or high trigs) as "One of more CVD enhancing factors" now in Table 2. We haven't put these people in a new category as it is not clear which category they should move up into (i.e.: not all enhancing factors are necessarily equivalent e.g.: the extra risk accrued by being obese, compared with the extra risk accrued from having high triglycerides) and whether these are additive if a person has more than one is also not clear. So the exact line you wanted in the table does not appear, but the new information allows you to estimate it, if for

	example you want to move all the people with one or more CVD enhancing factor from <10% up one category to 10-20% (but note this is not what is described in the WHO manual).
Citations number 22 and 25 are missing journal information.	Thanks for highlighting this, we have amended accordingly.

VERSION 2 – REVIEW

REVIEWER	Rosario Caruso IRCCS Policlinico San Donato
REVIEW RETURNED	03-Jul-2019

GENERAL COMMENTS	In my opinion, the response to the comments and the amendments in the manuscript are adequate.
--

REVIEWER	Pär Andersson White Linköping University, Sweden
REVIEW RETURNED	25-Jun-2019

GENERAL COMMENTS	Thank you for the new improved manuscript! My only remaining remark is that the conclusion in the abstract differs from the new conclusion in the discussion. The abstract still states that “CVD risk may be a lesser issue in slums than in other areas of LMICs cities” which cant be concluded because we do not know if the risk is different in the investigated slum areas than in Nairobi in general. The discussion states more correctly that “this study shows (insert: that) there is a low risk profile of CVD in this slum population in Nairobi, Kenya”
---

REVIEWER	Eleanor Axson Imperial College London, UK
REVIEW RETURNED	28-Jun-2019

GENERAL COMMENTS	This is an interesting paper making use of rare data. In the limitations paragraph, you mention cholesterol measures twice, I believe this is redundant.
--

VERSION 2 – AUTHOR RESPONSE

Response to comments by Reviewer 3

Reviewer: "My only remaining remark is that the conclusion in the abstract differs from the new conclusion in the discussion. The abstract still states that “CVD risk may be a lesser issue in slums

than in other areas of LMICs cities” which can’t be concluded because we do not know if the risk is different in the investigated slum areas than in Nairobi in general. The discussion states more correctly that “this study shows (insert: that) there is a low risk profile of CVD in this slum population in Nairobi, Kenya”

Thanks for pointing this out, we have amended the conclusion of the abstract to "This study shows that there is a low risk profile of CVD in this slum population in Nairobi, Kenya in comparison to results from application of multivariable risk prediction tools in other LMIC populations."

Response to comments by Reviewer 4

Reviewer: "In the limitations paragraph, you mention cholesterol measures twice, I believe this is redundant."

We have deleted "cholesterol (although optional) should be measured at two time points". Cholesterol is only now mentioned once in the limitations paragraph.